# Antimicrobial Activity, Genetic Relatedness, and Safety Assessment of Potential Probiotic Lactic Acid Bacteria Isolated from a Rearing Tank of Rotifers (*Brachionus plicatilis*) Used as Live Feed in Fish Larviculture

**DOI:** 10.3390/ani14101415

**Published:** 2024-05-09

**Authors:** Diogo Contente, Lara Díaz-Formoso, Javier Feito, Beatriz Gómez-Sala, Damián Costas, Pablo E. Hernández, Estefanía Muñoz-Atienza, Juan Borrero, Patrícia Poeta, Luis M. Cintas

**Affiliations:** 1Grupo de Seguridad y Calidad de los Alimentos por Bacterias Lácticas, Bacteriocinas y Probióticos (SEGA-BALBP), Sección Departamental de Nutrición y Ciencia de los Alimentos, Facultad de Veterinaria, Universidad Complutense de Madrid, 28040 Madrid, Spain; diogodas@ucm.es (D.C.); lardia01@ucm.es (L.D.-F.); ehernan@vet.ucm.es (P.E.H.); ematienza@vet.ucm.es (E.M.-A.); jborrero@ucm.es (J.B.); 2APC Microbiome Ireland, University College Cork, T12 YT20 Cork, Ireland; beatriz.gomezsala@teagasc.ie; 3Teagasc Food Research Centre, Moorepark, R93 XE12 Cork, Ireland; 4Centro de Investigación Mariña, Universidade de Vigo, Centro de Investigación Mariña (ECIMAT), 36331 Vigo, Spain; dcostas@ecimat.org; 5Microbiology and Antibiotic Resistance Team (MicroART), Department of Veterinary Sciences, Universidade de Trás-os-Montes e Alto Douro (UTAD), 5000-801 Vila Real, Portugal; ppoeta@utad.pt; 6CECAV-Veterinary and Animal Research Centre, University of Trás-os-Montes and Alto Douro (UTAD), 5000-801 Vila Real, Portugal; 7Associate Laboratory for Animal and Veterinary Science (AL4AnimalS), University of Trás-os-Montes and Alto Douro (UTAD), 5000-801 Vila Real, Portugal

**Keywords:** aquaculture, rotifers *(Brachionus plicatilis*), probiotics, Lactic Acid Bacteria (LAB), antimicrobial activity, in vitro safety assessment

## Abstract

**Simple Summary:**

Larviculture is one of the major bottlenecks in current aquaculture, where infectious diseases are responsible for populational collapses and large economic losses. Consequently, the main goal of this study was to evaluate the suitability of rearing tanks of rotifers (*Brachionus plicatilis*) used as the first live feed in turbot (*Scophthalmus maximus*, L.) larviculture as a new environmental niche for the isolation of bacterial probiotic candidates in the context of a sustainable aquaculture. Furthermore, the probiotic potential of four lactobacilli (two *Lacticaseibacillus paracasei* and two *Lactiplantibacillus plantarum*) isolated from a rotifer-rearing tank was evaluated.

**Abstract:**

Aquaculture is a rapidly expanding agri-food industry that faces substantial economic losses due to infectious disease outbreaks, such as bacterial infections. These outbreaks cause disruptions and high mortalities at various stages of the rearing process, especially in the larval stages. Probiotic bacteria are emerging as promising and sustainable alternative or complementary strategies to vaccination and the use of antibiotics in aquaculture. In this study, potential probiotic candidates for larviculture were isolated from a rotifer-rearing tank used as the first live feed for turbot larvae. Two *Lacticaseibacillus paracasei* and two *Lactiplantibacillus plantarum* isolates were selected for further characterization due to their wide and strong antimicrobial activity against several ichthyopathogens, both Gram-positive and Gram-negative. An extensive in vitro safety assessment of these four isolates revealed the absence of harmful traits, such as acquired antimicrobial resistance and other virulence factors (i.e., hemolytic and gelatinase activities, bile salt deconjugation, and mucin degradation, as well as PCR detection of biogenic amine production). Moreover, Enterobacterial Repetitive Intergenic Consensus-PCR (ERIC-PCR) analyses unveiled their genetic relatedness, revealing two divergent clusters within each species. To our knowledge, this work reports for the first time the isolation and characterization of Lactic Acid Bacteria (LAB) with potential use as probiotics in aquaculture from rotifer-rearing tanks, which have the potential to optimize turbot larviculture and to introduce novel microbial management approaches for a sustainable aquaculture.

## 1. Introduction

Aquaculture, which is described by the Food and Agriculture Organization (FAO) as the farming of aquatic organisms (including fish, mollusks, crustaceans, and plants) with variable levels of human intervention, is the fastest-growing agri-food sector worldwide [1]. In recent years, aquaculture intensification has led to a growing interest in fish nutrition and the development of feeding options, which, currently, in finfish husbandry, mostly rely on algae, live feed (such as *Artemia* spp., copepods, and rotifers), and commercially available fish feeds [2].

Infectious disease outbreaks are a critical limiting factor for aquaculture, being associated with high mortality rates and, therefore, important economic losses. Bacterial infections, which represent around 55% of the total reported outbreaks, enter the rearing systems via supplied water, broodstock, human handling, or the live feed. This is of special importance during the fish larvae rearing phase, as larvae are exposed to high organic load and bacterial concentrations, in nutrient-rich waters, fed on live feed, and with underdeveloped gut microbiome and immune systems [2,3,4,5,6,7]. Larvae populational crashes can occur during the first live feed phases, but also during imbalances between the larvae and microbial communities, in which opportunistic and pathogenic bacteria proliferate [2,7]. Therefore, controlling the microbial communities and the balance associated with the aquatic rearing systems are deemed as fundamental. Traditionally, this control has been achieved through the disinfection of the rearing tanks, sterilization of the supplied water, and prophylactic use of antibiotics [2,5,6,7,8,9,10,11]. Nevertheless, there is a growing interest within the aquaculture industry for sustainable alternative or complementary strategies to the use of antibiotics, due to their harmful effects on animal, human, and environmental health [5,12]. This interest has been mostly directed towards vaccination and the use of probiotics. However, due to the immature state of the larvae immune system, which renders vaccination less effective and costlier, the interest for probiotic bacteria has been growing [2,12,13,14].

In aquaculture, probiotics can be defined as live micro-organisms, usually bacteria or yeasts, that, when administered through feed or the rearing environment, confer beneficial effects to the host such as protection against ichthyopathogens, the stimulation of the immune system, competition for adhesion sites in the mucosa, improved tolerance to stress, host nutrition enhancement, and water quality improvement [14,15,16,17,18]. Usually, both the host and the rearing environment have been regarded as the most suitable sources for the isolation of probiotic candidates for larviculture. In this context, the larviculture of finfish, such as turbot (*Scophthalmus maximus* L.), has been, for decades, relying on rotifers (*Brachionus plicatilis*) as the live feed. Rotifers are small-sized live preys, and non-selective filters, that are crucial for delivering essential nutrients and probiotics to underdeveloped fish larvae. Currently, all rotifers are enriched before being supplied to fish larvae, not only to meet nutritional demands, but also to enhance growth, survival rates, and stress tolerance, and promote microbial diversity [19,20,21,22,23,24]. Nevertheless, the rearing tank of rotifer cultures has been recognized as an optimal environment for the growth of pathogenic and opportunistic bacteria, such as *Aeromonas* spp. and *Vibrio* spp., as rotifers are grown in high-populational-density tanks, with a high load of organic matter, and fed mainly on microalgae and yeasts. Hence, strategies to disinfect and clean rotifers of unwanted opportunistic and pathogenic bacteria are a topic of growing interest. However, such strategies have often had a lethal result on the rotifers or yielded inconclusive results [22,23,24]. On the other hand, several authors have previously demonstrated the benefits of enriching rotifers with probiotic bacteria in finfish larviculture systems. The reported beneficial effects include enhanced larvae growth [25], a positive influence in early gut colonization [3,26,27], and protective effects of turbot larvae against, amongst others, *Vibrio anguillarum* infections [4]. Therefore, the interaction and combination between probiotic bacteria, rotifers, and finfish larvae has been shown to be not only a promising approach for the optimization of this demanding stage of modern aquaculture, but also a possible source for new antimicrobial community control strategies in rotifer farming.

The objectives of the present study were (i) the isolation of culturable microbiota from a rearing tank of rotifers used as the first live feed for the larviculture of turbot in northwestern Spain; (ii) the evaluation of the antimicrobial activity of the bacterial isolates against several Gram-positive- and Gram-negative-relevant ichthyopathogens; (iii) the taxonomic identification and genetic relatedness of selected isolates; and (iv) the in vitro safety assessment of the most promising probiotic candidates for turbot farming as an alternative or complementary strategy to antibiotic therapy and vaccination for disease prevention and rotifer enrichment.

## 2. Materials and Methods

### 2.1. Bacterial Isolation, Sampling, and Growth Conditions

The bacterial isolates used in the present study were obtained from a rearing tank of rotifers used as live feed in an experimental turbot farm located in Galicia (northwestern Spain). Samples were collected from three different origins: (i) tank vegetation (biofilm, BF), (ii) tank water with rotifers (RT), and (iii) tank filtered-water (WT). Subsequently, the samples from the three different origins were ten-fold diluted in sterile peptone water (Oxoid, Basingstoke, UK), pour-plated onto de Man, Rogosa, and Sharpe (MRS, Oxoid, UK) agar (1.5%, *w*/*v*) (Scharlab, Barcelona, Spain) plates, and further incubated aerobically at 30 °C for 24–48 h.

### 2.2. Direct Antimicrobial Activity

A total of 45 isolates (15 from each different origin) were selected based on morphological characteristics, and then assayed for direct antimicrobial activity by a Stab-On-Agar Test (SOAT), as previously described by Cintas et al. [28], against eight Gram-negative and three Gram-positive ichthyopathogens. The Gram-negative ichthyopathogens *Aeromonas hydrophila* CECT839, *A. hydrophila* CECT5734, *A. salmonicida* CECT894, *A. salmonicida* CECT4237, *A. salmonicida* CLFP23, *Edwardsiella tarda* CECT886, and *Yersinia ruckeri* LMG3279 were aerobically grown in Tryptone Soya Broth (TSB, Oxoid, UK) at 25 °C overnight, while *Vibrio anguillarum* CECT4344 was aerobically grown in TSB supplemented with NaCl (1.5%, *w*/*v*) (Thermo Scientific, Waltham, MA, USA) at 25 °C overnight. The Gram-positive ichthyopathogens *Lactococcus garvieae* CF00021 and *L. garvieae* CLG4 were aerobically grown in MRS broth at 30 °C overnight, while *Streptococcus parauberis* was grown in Brain Heart Infusion broth (BHI, Oxoid, UK) at 37 °C overnight. The isolates showing antimicrobial activity (absence of visible microbial growth around the stabbed cultures) against, at least, one of the tested pathogens were selected and stored in their corresponding culture media containing 15% (*v/v*) glycerol (Thermo Scientific) at –20 and –80 °C, until further use.

### 2.3. Taxonomic Identification of Selected Isolates

Twelve different isolates (with each different origin being represented), selected due to their direct antimicrobial activity, were taxonomically identified by DNA partial sequencing of the PCR-amplified gene encoding the 16S rRNA subunit (*16S rDNA*). PCR-amplifications were performed from total bacterial DNA, which was purified using the InstaGene Matrix (BioRad Laboratories Inc., Hercules, CA, USA) in 50 μL reaction mixtures with 5–50 ng of purified DNA template, 1 μL of each 7 × 10^−5^ mol/L primer, and 25 μL of Mytaq Mix (Bioline Reagents, Ltd., London, UK). PCR cycling conditions were conducted in a MJ Mini Gradient Thermal Cycler (BioRad Laboratories, Inc.) as follows: one initial denaturation step at 95 °C (1 min), 35 cycles of denaturation–annealing–extension (95 °C for 15 s, 55 °C for 15 s, and 72 °C for 10 s, respectively), and a final extension step at 72 °C (4 min). After agarose (1.5%, *w*/*v*) (Pronadisa, Madrid, Spain) gel electrophoresis, dyed with GelRed Nucleic Acid Gel Stain (Biotium, Inc., Fremont, CA, USA), resulting bands were visualized in a ChemiDoc Imaging System (BioRad Laboratories, Inc.). The oligonucleotide primers used for PCR amplification of *16S rDNA* were plb16 (50′-AGAGTTTGATCCTGGCTCAG-3′) and mlb16 (5′-GGCTGCTGGCACGTAGTTAG-3′) [29]. HyperLadder II (Bioline GmbH, Luckenwalde, Germany) was used as molecular size marker. The amplicons were purified by using the NucleoSpin Extract II kit (Macherey & Nagel, Düren, Germany) and the DNA strands were sequenced at the Unidad de Genómica (Parque Científico de Madrid, Facultad de Ciencias Biológicas, Universidad Complutense de Madrid, Spain). Analysis of *16SrDNA* sequences was performed with the BLAST program available at the National Center for Biotechnology Information (NCBI; blast.ncbi.nlm.nih.gov, accessed on 17 March 2024).

### 2.4. Molecular Typing and Genetic Relatedness: Enterobacterial Repetitive Intergenic Consensus-PCR (ERIC-PCR)

ERIC-PCR analysis of the 12 isolates was carried out by using primers ERIC-1R (5′-ATGTAAGCTCCTGGGGGGATTCAC-3′) and ERIC-2 (5′-AAGTAAGTGACTGGGGG GTGAGCG-3′) as previously described [21]. Then, 50 μL PCR-reaction mixtures were prepared with 25 μL of MyTaq Mix (Bioline Reagents, Ltd., London, UK), 0.7 µM of each primer, 5–50 ng of purified DNA, 3 μM of MgCl_2_, and 19 µL of molecular-biology-grade water. PCR mixtures were subjected to an initial denaturation (95 °C, 1 min), 35 cycles of denaturation–annealing–elongation (95 °C, 15 s; 46 °C, 15 s; and 72 °C, 10 s), and a final elongation (72 °C, 4 min) in a thermal cycler (Eppendorf, Hamburg, Germany). The amplification products were gel-electrophoresed at 90 V for 60 min in an agarose gel (1.5% *w*/*v*) (Pronadisa), dyed with GelRed Nucleic Acid Gel Stain (Biotium, Inc.), in an electrophoresis chamber (BioRad Laboratories, Inc.), and subsequently visualized using the ChemiDoc Imaging System (BioRad Laboratories, Inc.), with HyperLadder 100 bp (Bioline Reagents, Ltd.) as molecular weight marker. ERIC-PCR type analysis, clustering, and dendrogram construction were performed by using the Phoretix v.5.0 software (Nonlinear Dynamics Ltd., Newcastle-upon-Tyne, UK).

### 2.5. In Vitro Safety Assessment

#### 2.5.1. Antibiotic Susceptibility Testing

The Minimum Inhibitory Concentration (MIC) of 12 antibiotics against the selected probiotic candidates was determined by a broth microdilution test [30], with slight modifications [21]. The antibiotics were chosen according to the European Food Safety Authority (EFSA, Parma, Italy) Panel on Additives and Products or Substances used in Animal Feed (FEEDAP) guidelines on the guidance on the characterization of micro-organisms used as feed additives or as production organisms [31]. Additionally, some of the antibiotics frequently used in aquaculture, which were not listed by the FEEDAP guidelines, such as amoxicillin, florfenicol, oxytetracycline, and trimethoprim-sulfamethoxazole, were also included in the assay [32,33,34]. The breakpoints utilized for most of the antibiotics were the ones established by the EFSA guidelines, while the breakpoints consulted for the antibiotics regularly used in aquaculture were the ones established by the Clinical & Laboratory Standards Institute (CLSI, Wayne, PA, USA). The antibiotics tested on this assay were: amoxicillin (8–0.006 μg/mL), ampicillin (16–0.25 μg/mL), chloramphenicol (64–1 μg/mL), clindamycin (16–0.25 μg/mL), erythromycin (16–0.25 μg/mL), florfenicol (8–0.006 μg/mL), gentamicin (32–0.5 μg/mL), kanamycin (128–2 μg/mL), oxytetracycline (4–0.03 μg/mL), streptomycin (64–1 μg/mL), sulfamethoxazole/trimethoprim (76/4–0.6/0.03 μg/mL), and tetracycline (32–0.5 μg/mL). Strains were considered resistant when their MIC for one specific antimicrobial agent was higher than the respective cut-off value. Quality control was performed using the strains *Enterococcus faecalis* CECT795, and *Staphylococcus aureus* CECT794.

#### 2.5.2. Hemolytic and Gelatinase Activities

The method firstly described by Eaton and Gasson [35] and modified by Muñoz-Atienza et al. [36] was used to study the hemolytic and gelatinase activities of the selected probiotic candidates. The α- and β-hemolysis were revealed by the appearance of green-like halos and clear zones, respectively, around and below the cultures. *Streptococcus pneumoniae* FQ6 and *E. faecalis* SDP10 were used as positive controls for α- and β-hemolysis, respectively. On the other hand, the presence of a cloudy halo around the colonies was interpreted as gelatin hydrolysis, for which *E. faecalis* P4 was used as a positive control.

#### 2.5.3. Bile Salt Deconjugation

The ability of the selected probiotic candidates to deconjugate bile salts (taurocholate and taurodeoxycholate) was evaluated through the method previously described by Noriega et al. [37]. The appearance of an opaque bubbly halo around the cultures was indicative of a positive result. Fresh feces from a healthy adult dairy cow (*Bos taurus*) was used as positive controls.

#### 2.5.4. Mucin Degradation

The ability of the selected probiotic candidates to degrade mucin was analyzed following the method previously described by Zhou et al. [38]. The appearance of discolored halos surrounding the cultures was interpreted as a positive result. Fresh feces from a healthy adult dairy cow (*Bos taurus*) was used as positive controls.

#### 2.5.5. Biogenic Amine Production PCR-Detection

The isolated DNA was subjected to PCR amplifications to detect the presence of the histidine decarboxylase (*hdc*), tyrosine decarboxylase (*tdc*), and ornithine decarboxylase (*odc*) genes by using the primers CL1-JV17HC, TD2-TD5, and 3–16, respectively, as previously described [39,40,41], PCR products were visualized as described above. *Lactobacillus* sp. 30A and *Enterococcus faecium* L50 were used as positive and negative controls for *hdc* and *odc*, and *tdc,* respectively [42].

### 2.6. Biofilm Formation Assays

Microbial biofilm formation assays were performed as described by Silva et al. [43]. *S. aureus* ATCC^®^ 25923 was used as a positive control, and the results expressed as percentages relative to those of this reference strain. Fresh medium without bacterial inoculum was used as a negative control. All experiments had seven technical replicates and were performed in triplicate. Biofilm mass was quantified using the modified version described in [43] of the Crystal Violet (CV) Staining method firstly described by Peeters et al. [44].

### 2.7. Statistical Analysis

Data curation, statistical analyses, and graphical representations were performed using the GraphPad Prism 8 software (GraphPad Software, San Diego, CA, USA). All data were verified for normal distribution using the Shapiro–Wilk test and transformed when required by Napierian logarithm. Statistical analyses were then performed using the Welch’s test for 24–48 h comparisons between the same strain, and one-way analysis of variance (ANOVA) when comparing data between different isolates, followed by Tukey’s post hoc tests, when appropriate.

## 3. Results and Discussion

### 3.1. Antimicrobial Activity, Taxonomic Identification, and Molecular Typing of Five Lacticaseibacillus Paracasei and Seven Lactiplantibacillus Plantarum Isolates

The 45 isolates from the rearing tank of rotifers used as the first live feed in turbot larviculture exerted direct antimicrobial activity against, at least, one of the 11 ichthyopathogens used as indicator micro-organisms (Table 1). Moreover, 30 isolates (*ca*., 66.7%) demonstrated direct antimicrobial activity towards five or more of them (Figure 1). Moreover, three isolates (*ca.*, 6.7%) showed antimicrobial activity against seven indicator micro-organisms.

Interestingly, the bacterial strains isolated from the rotifer tank inhibited ichthyopathogens of paramount importance in turbot larviculture and mariculture, such as *A. salmonicida* [45], *E. tarda* [46], *V. anguillarum* [47], and *St. parauberis* [48]. Although *A. salmonicida* CECT4237 was sensitive to all the LAB isolates, none of them inhibited *A. salmonicida* CECT894. This strain-specific antimicrobial sensitiveness has been previously reported for different antimicrobial mechanisms, such as copper and silver nanoparticles, contact-independent bactericidal compounds, or even lantibiotics, which can partially justify our results [49,50]. Classical furunculosis, which is caused by *A. salmonicida*, is one of most impactful bacterial diseases in aquaculture, causing an important economic impact, as this species generally causes an acute fatal hemorrhagic septicemic disease [46]. Furthermore, none of the isolates inhibited *A. hydrophila* CECT839, *A. hydrophila* CECT5734, nor *Y. ruckeri* LMG3279. Moreover, approximately 26.7% of the isolates demonstrated a wide antimicrobial activity, which was considered as the inhibition of six or more different ichthyopathogens.

The antimicrobial activity exerted by LAB can be due to several mechanisms, including the competition for nutrients and production of organic acids, as well as other antimicrobial substances, such as ethanol, hydrogen peroxide, and ribosomally synthesized peptides or proteins (i.e., bacteriocins) [16,51,52]. The 12 isolates (26.7%) that exerted direct antimicrobial activity against, at least, six indicator micro-organisms were taxonomically identified as *Lactiplantibacillus plantarum* (seven, 58%), and as *Lacticaseibacillus paracasei* (five, 42%). Out of the seven *Lp. plantarum*, three were isolated from the tank water containing rotifers (RT) (*ca.*, 42%), two from the tank vegetation (biofilm, BF) (*ca.*, 29%), and the remaining two from tank filtered-water without rotifers (WT) (*ca.*, 29%). On the other hand, two of the *Lc. paracasei* were isolated from the tank filtered-water without rotifers (WT) (40%), two from the tank water with rotifers (RT) (40%), and one from the tank vegetation (biofilm, BF) (20%).

An analysis of the antimicrobial spectrum exerted by the 12 taxonomically identified LAB revealed that, out of the five *Lc. paracasei*, three (*ca.*, 60.0%) inhibited the growth of six indicator micro-organisms, while the remaining two (*ca.*, 40.0%) were active against seven of them. Regarding the seven *Lp. plantarum*, six (*ca.*, 85.7%) inhibited the growth of six indicator micro-organisms, whilst only one isolate (*ca.*, 14.3%) was able to inhibit all the indicators. Furthermore, three *Lc. paracasei* (*ca.*, 60.0%) inhibited two out of the three Gram-positive ichthyopathogens, whilst two (*ca.,* 40.0%) inhibited all of them. On the other hand, six *Lp. plantarum* (*ca.*, 85.7%) inhibited two Gram-positive ichthyopathogens, whereas only one (*ca.*, 14.3%) exerted antimicrobial activity against all the Gram-positive indicators. Regarding the antimicrobial spectrum of the 12 taxonomically identified LAB towards the eight Gram-negative ichthyopathogens, all the *Lc. paracasei* inhibited four of them, (50.0%), whereas five (*ca.*, 71.4%) and two (*ca.*, 28.6%) *Lp. plantarum* were active against three and four Gram-negative ichthyopathogens, respectively.

Interestingly, *Lp. plantarum* strains have often been described as suitable probiotic candidates for aquaculture due to their beneficial effects on a wide array of aquatic target-species, including crustaceans, such as marron (*Cherax cainii*) [53], freshwater fish, such as common carp (*Cyprinus carpio*) [54], and marine species, such as turbot [55]. Furthermore, some authors have reported the antimicrobial activity and protection effects of *Lp. plantarum* strains against Gram-negative ichthyopathogens, such as *A. salmonicida* [56], *A. hydrophila* [57], and *Vibrio harveyi* [58] Similarly, *Lc. paracasei* strains have previously been associated with probiotic effects in several farmed aquatic species, such as Nile tilapia [59], and white-leg shrimp (*Penaeus vannamei*) [60], as well as by demonstrating antimicrobial activity against Gram-negative aquatic pathogens such as *Vibrio parahaemolyticus* [61]. Moreover, the use of *Lc. paracasei* strains has been associated with improved fish feed utilization and nutrient bioavailability [62].

The genetic relatedness of the five *Lc. paracasei* and seven *Lp. plantarum* isolates was accomplished by ERIC-PCR, a molecular fingerprinting method widely and successfully used for bacterial typing and epidemiological studies to determine, for instance, the genetic relatedness between isolates from different origins [43,44]. In this regard, four of the *Lc. paracasei* were grouped into a single main cluster (similarity ≥ 75%), showing a 50% similarity to the remaining isolate (Figure 2a). As for the *Lp. plantarum* isolates, six of them were grouped in a main cluster (100% similarity), showing a 80% similarity to the main cluster (Figure 2b). Notably, isolates representing all the three tank origins were present in the main clusters identified for both species.

Interestingly, isolates within the same ERIC-PCR cluster did not show the same antimicrobial spectrum (Table 1). For instance, within the main *Lp. plantarum* cluster, not all isolates inhibited *E. tarda*. These results and observations are somewhat in agreement with previous studies. In this regard, ERIC-PCR was shown to be an easy and rapid strategy for the molecular fingerprinting of LAB, but, also, as observed by other authors, similarities observed in DNA fingerprinting techniques do not always directly correlate with similar phenotypic characteristics [42,63]. Furthermore, ERIC-PCR analyses suggested that the broad antimicrobial spectrum of *Lc. paracasei* and *Lp. plantarum* strains could be considered an advantage when evaluating them as potential probiotics for aquaculture, as these strains are native to both the host and the environment. Therefore, this observation maximizes the chances of these strains not only surviving and thriving spontaneously, but also performing their beneficial physiological and metabolic functions in these ecological niches [64,65].

### 3.2. In Vitro Safety Assessment of the Four Selected LAB Strains

The use of probiotics introduced into an aquatic environment may interfere with and modify the microbial ecology and safety of aquatic hosts, sediments, and the associated environment [65,66]. Therefore, the in vitro safety assessment of candidate probiotics is of paramount importance. In this work, two *Lc. paracasei* (BF3 and RT4) and two *Lp. plantarum* (BF12 and WT12) strains isolated from a rotifer-rearing tank used as the first live feed for turbot larvae, representing each of the four identified ERIC-PCR clusters, were further selected, based on their broad antimicrobial spectrum and genetic relatedness, for an extensive in vitro safety assessment.

According to the microbiological breakpoints established for *Lc. paracasei* and *Lp. plantarum* by the FEEDAP document for ampicillin, chloramphenicol, clindamycin, erythromycin, gentamicin, kanamycin, streptomycin, and tetracycline, as well as those established by CLSI for amoxicillin, florfenicol, oxytetracycline and sulfamethoxazole/trimethoprim, the four selected lactobacilli were found to be susceptible to all tested antibiotics (results not shown). Furthermore, none of these strains showed resistance to four antibiotics, namely, amoxicillin, florfenicol, oxytetracycline, and sulfamethoxazole/trimethoprim, which are not included in the EFSA FEEDAP requirements [31,67], but which do belong to a group of antimicrobials often used in aquaculture [32,33,34]. In this context, the overuse of antibiotics in aquaculture poses a real threat to human and veterinary medicine, and, due to the characteristics of fish farms, which tend to have fewer barriers than other land-farmed species, it also poses a major threat to adjacent ecosystems. In this respect, several studies have demonstrated the existence of antibiotic resistance genes in sediments and waters from fish farm pens and ponds, which may be responsible for changes in the resistome of large seas, such as those recently found in the Baltic Sea, or even in the intercontinental spread of antibiotic-resistant *Aeromonas* spp. [68,69,70].

Interestingly, none of the four selected lactobacilli exerted hemolytic or gelatinase activities, degraded mucin, or showed bile-deconjugating activity. In this context, hemolytic activity, causing membrane damage, cell lysis, and the destruction of neighboring cells and tissues, is commonly associated with pathogenic bacteria, and thus regarded as a relevant virulence factor [71]. Similarly, gelatinase activity has been associated with bacterial pathogenicity and, therefore, also regarded as a virulence factor [45,72]. Moreover, the excessive deconjugation of bile salts has been considered as an unfavorable trait in animal husbandry, as unconjugated bile acids are less efficient than conjugated ones in the emulsification of dietary lipids. In addition, the formation of micelles, lipid digestion, and absorption of fatty acids and monoglycerides could be compromised by deconjugated bile salts. Likewise, the degradation of mucin is seen as a detrimental characteristic, as it can facilitate the invasion of pathogenic bacteria through the mucosal barrier into extra-intestinal tissues [36,45].

On the other hand, PCR analyses revealed that none of the four selected probiotic LAB candidates produced the decarboxylase enzymes required to form biogenic amines, namely, histamine, tyramine, and putrescine, which are the main causative agents of seafood-borne illnesses. In this respect, histamine is of particular relevance, since it is associated with scombroid poisoning, while tyramine can be associated with hypertensive episodes and migraines. Moreover, putrescine can potentiate the effects of histamine, as well as reacting with nitrite to form carcinogenic nitrosamines [45,73].

### 3.3. Biofilm Formation by the Four Selected LAB Strains

The two *Lp. plantarum* strains showed a greater ability to form biofilms, both at 24 and 48 h, compared to the control strain and the *Lc. paracasei* isolates (Figure 3). Interestingly, *Lp. plantarum* BF12, which was isolated from the tank biofilm/vegetation, proved to be the strain with the highest ability to attach and form biofilms. Furthermore, when comparing the biofilm formation capacity after 24 and 48 h, the four lactobacilli showed a statistically significant higher biofilm formation capacity at 48 h, excepting *Lc. paracasei* RT4, which was the strain that not only showed the lowest biofilm formation capacity, but also no statistically significant difference at both temperatures.

The cell surface properties shown by biofilm-producing beneficial bacteria have been recently proposed as a new probiotic mechanism [74,75]. In this regard, there has been growing interest in the biofilm-forming capabilities of bacteria belonging to the former genus *Lactobacillus*, including *Lc. paracasei* and *Lp. plantarum* species [76,77,78,79]. Therefore, biofilm formation is now considered not only a new antagonistic strategy to control the growth of unwanted bacteria, but also a mechanism employed by probiotic bacteria to withstand stress conditions, such as severe pH changes and nutrient starvation [75,76,77,78,79,80]. In this regard, our study demonstrates a higher biofilm-forming capacity of *Lp. plantarum* compared to *Lc. paracasei*, which is somewhat in agreement with previous reports, which have detailed the beneficial cell surface properties and capabilities of *Lp. plantarum* species. Finally, our results reveal that a 48 h incubation is the optimal period for biofilm formation by the two potential probiotic *Lp. plantarum* strains, which is in agreement with previous results on this species [80].

## 4. Conclusions

Aquaculture systems are extremely nutrient-rich environments with high levels of microbial instability, which favor the growth of opportunistic fish pathogenic bacteria, and, therefore, often causing microbial interference detrimental to fish larvae. Since larviculture heavily relies on feeding zooplankton (e.g., rotifers, copepods, cladocerans, and *Artemia* spp.), new sustainable biocontrol approaches, such as the use of probiotic bacteria (e.g., *Lp. plantarum*), are emerging as promising alternative or complementary biocontrol strategies to vaccination and antibiotic use, for this critical phase of fish farming systems. To our knowledge, the present work reports, for the first time, the isolation of several probiotic LAB candidates from rotifer-rearing tanks (tank vegetation/biofilms, tank water, and filtered tank water) used as the first live feed in turbot larviculture, and suggests the suitability of these ecological niches for the isolation of probiotic candidates. The broad and strong antimicrobial spectrum shown by the four selected *Lc. paracasei* (BF3 and RT4) and *Lp. plantarum* (BF1 and WT12) strains against several Gram-positive and Gram-negative ichthyopathogens of special relevance in turbot aquaculture, as well as against other fish pathogens, together with their lack of safety concerns, allows us to recognize these LAB strains as promising probiotic candidates to be used as a new biocontrol strategy in aquaculture. However, further studies must be conducted to fully understand and unravel the mechanisms responsible for their broad antimicrobial activity against ichthyopathogens, as well as to identify their probiotic traits by Whole-Genome Sequencing (WGS) and in silico analysis, and to evaluate in vivo their suitability for use as probiotics in the context of a sustainable aquaculture.

## Figures and Tables

**Figure 1 animals-14-01415-f001:**
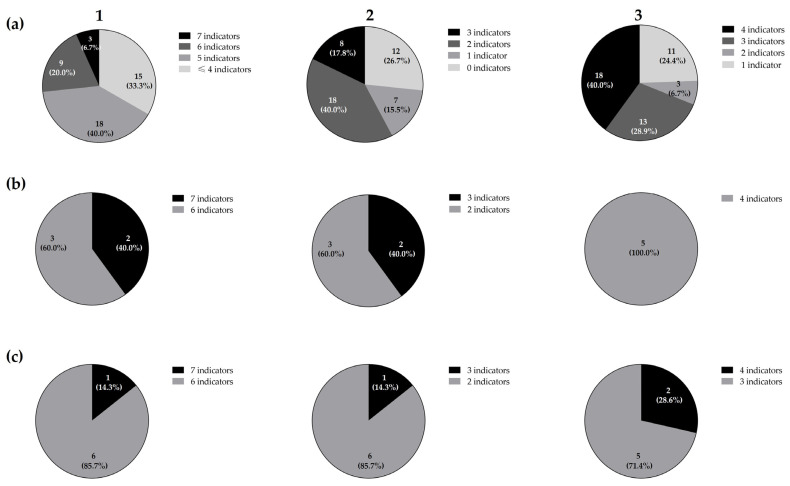
Distribution of the total 45 LAB isolates (**a**), five *Lc. paracasei* (**b**), and seven *Lb. plantarum* (**c**) isolates from the rearing tank of rotifers according to their direct antimicrobial activity spectrum against the 11 total (**1**), three Gram-positive (**2**), and eight Gram-negative (**3**) ichthyopathogens by a SOAT.

**Figure 2 animals-14-01415-f002:**
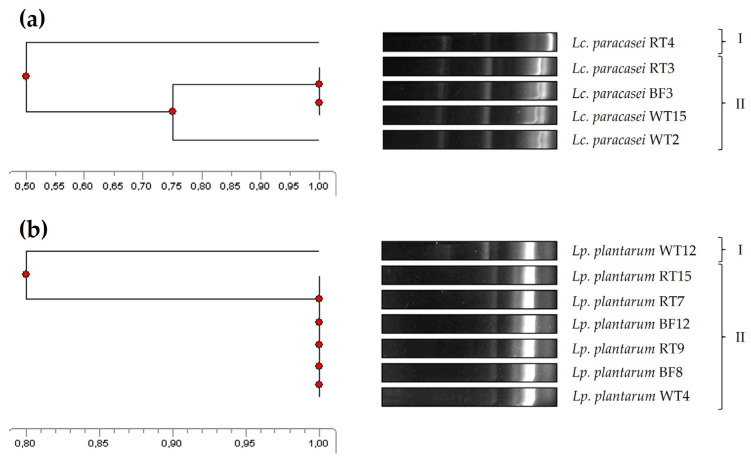
Phylogenetic relatedness of the *Lc. paracasei* (**a**) and *Lp. plantarum* (**b**) isolates from the rearing tank of rotifers, based on their ERIC-PCR patterns.

**Figure 3 animals-14-01415-f003:**
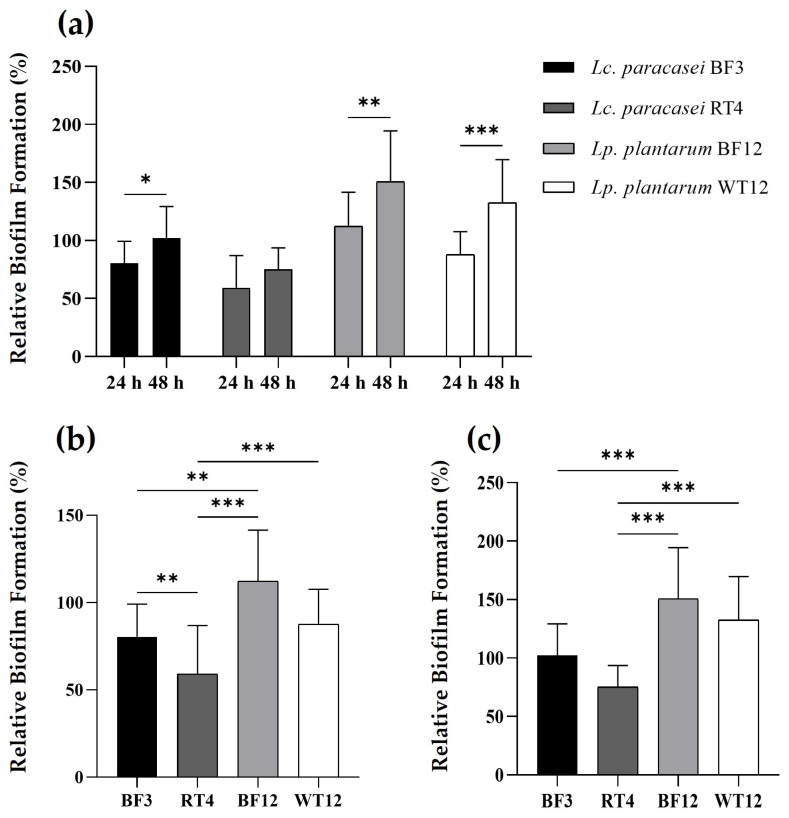
Time-dependent comparison of relative biofilm formation by *Lc. paracasei* BF3, *Lc. paracasei* RT4, *Lp. plantarum* BF12, and *Lp. plantarum* WT12, at 24 and 48 h (**a**). Relative biofilm formation comparison between LAB strains at 24 h (**b**), and 48 h (**c**). Results are expressed as percentages relative to those of the reference strain (*S. aureus* ATCC^®^ 25923). The asterisks indicate levels significantly different between bacterial treatments (* *p* ≤ 0.05, ** *p* ≤ 0.01, and *** *p* ≤ 0.001).

**Table 1 animals-14-01415-t001:** Origin and direct antimicrobial activity ^a^ of the 45 isolates from the rearing tank of rotifers against 11 ichthyopathogens.

Isolates	Indicator Micro-Organisms										
	*A. hydrophila* CECT839	*A. hydrophila* CECT5734	*A. salmonicida* CECT4237	*A. salmonicida* CECT894	*A. salmonicida* CLFP23	*E. tarda* CECT886	*V. anguillarum* CECT4344	*Y. ruckeri* LMG3279	*L. garvieae* CF00021	*L. garvieae* CLG4	*St. parauberis* LMG225
**Biofilm (BF)**											
BF1	-	-	+	-	-	-	-	-	-	-	-
BF2	-	-	+	-	-	-	-	-	-	-	-
BF3	-	-	+	-	+	+	+++	-	+	+	++
BF4	-	-	+	-	-	-	-	-	-	-	-
BF5	-	-	+	-	-	-	-	-	-	-	-
BF6	-	-	+	-	-	-	-	-	-	-	-
BF7	-	-	+	-	-	-	-	-	-	-	-
BF8	-	-	+	-	-	+	+	-	++	+	+
BF9	-	-	+	-	-	-	+	-	-	-	-
BF10	-	-	+	-	-	-	-	-	-	-	-
BF11	-	-	+	-	-	-	-	-	-	-	-
BF12	-	-	+	-	+++	+	+++	-	-	+	++
BF13	-	-	+	-	-	-	-	-	+	-	-
BF14	-	-	+	-	-	-	-	-	-	-	-
BF15	-	-	+	-	+	-	+	-	+	-	+
**Tank water with rotifers (RT)**											
RT1	-	-	+	-	+	-	+	-	+	+	-
RT2	-	-	+	-	+	-	+	-	-	+	+
RT3	-	-	+	-	+	+	++	-	-	++	+
RT4	-	-	+	-	+++	+	++	-	++	+++	++
RT5	-	-	+	-	-	-	+	-	-	-	+
RT6	-	-	+	-	-	-	-	-	-	-	-
RT7	-	-	+	-	+	-	+	-	+	+	+
RT8	-	-	+	-	+	-	+	-	-	+	-
RT9	-	-	+	-	++	-	+	-	+	+	+
RT10	-	-	+	-	+	-	-	-	-	-	-
RT11	-	-	+	-	+	-	+	-	-	+	+
RT12	-	-	+	-	+	-	+	-	-	+	+
RT13	-	-	+	-	+	-	+	-	-	+	+
RT14	-	-	+	-	+	-	+	-	+	+	-
RT15	-	-	+	-	++	-	+	-	+	++	+
**Tank water without rotifers (WT)**											
WT1	-	-	+	-	+	+	+	-	-	+	-
WT2	-	-	+	-	+	+	++	-	-	++	+
WT3	-	-	+	-	-	+	+	-	-	+	+
WT4	-	-	+	-	-	+	++	-	+	++	+
WT5	-	-	+	-	+	+	+	-	-	+	-
WT6	-	-	+	-	+	+	+	-	-	+	-
WT7	-	-	+	-	+	+	+	-	-	+	-
WT8	-	-	+	-	+	+	++	-	-	++	-
WT9	-	-	+	-	-	+	+	-	+	+	-
WT10	-	-	+	-	+	+	+	-	-	+	-
WT11	-	-	+	-	+	+	+	-	-	+	-
WT12	-	-	+	-	+++	++	++	-	+	++	++
WT13	-	-	+	-	+	+	+	-	-	+	-
WT14	-	-	+	-	+	+	+	-	-	+	-
WT15	-	-	+	-	++	+	++	-	-	+	+

^a^ The direct antimicrobial activity as determined by a SOAT, and the scores reflect the ranges of growth inhibition (diameter of the inhibition zones in mm): -, no inhibition; +, <5 mm; ++, ≥5–10 mm; +++, ≥10 mm.

## Data Availability

The data are available upon request from the corresponding authors. The prokaryotic 16S rRNA gene sequences have been deposited at DDBJ/ENA/GenBank under the accession numbers PP707637 (*Lc. paracasei* BF3), PP707192 (*Lc. paracasei* RT3) PP707591 (*Lc. paracasei* RT4), PP707590 (*Lc. paracasei* WT2), PP707592 (*Lc. paracasei* WT15), PP707194 (*Lp. plantarum* BF8), PP724700 (*Lp. plantarum* BF129, PP707089 (*Lp. plantarum* RT7), PP707090 (*Lp. plantarum* RT9), PP707085 (*Lp. plantarum* RT15), PP707620 (*Lp. plantarum* WT4), and PP707622 (*Lp. plantarum* WT12).

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
