# Peer review of "Antimicrobial Activity, Genetic Relatedness, and Safety Assessment of Potential Probiotic Lactic Acid Bacteria Isolated from a Rearing Tank of Rotifers (Brachionus plicatilis) Used as Live Feed in Fish Larviculture"

_animals, 2024, doi:10.3390/ani14101415_

Round 1
Reviewer 1 Report
Comments and Suggestions for Authors
In this article, the authors isolated 45 potential probiotic candidates from a rotifer rearing tank, examined their antimicrobial activity against several ichthyopathogens, and selected 2 Lacticaseibacillus paracasei and 2 Lactiplantibacillus plantarum strains. They further examined these 4 strains by in vitro safety assessment experiments.
The experiments were well-designed, and results contain several useful information on probiotics in aquaculture. The reviewer, however, found several concerns which should be addressed before publication as follows.
Major concern
The accession numbers of the 16S rRNA should be provided in “2.3. Taxonomic Identification of Selected Isolates” of the Materials and Methods section, because it is mentioned in Instructions for Authors of Animals that “Accession numbers of RNA, DNA and protein sequences used in the manuscript should be provided in the Materials and Methods section.”
The reviewer believes that this paper should not be published before the author provide the accession numbers of the 16S rRNA.
Figure 3
What are the Y-axis values?
In line 252 it says “S. aureus ATCC® 25923 was included as a positive control”, but no data on S. aureus ATCC® 25923 is shown in Figure 3. The reviewer thinks it is better to express the results as the relative values to that of positive control.
Minor points
119 to 121
Briefly, the 45 isolates were stabbed onto MRS agar plates and incubated at 30°C for 5 h, and then, 40 mL of the corresponding soft agar (0.8%, w/v) medium containing about 1×105 CFU/mL of the pathogen strain was poured onto the plates.
Please indicate the size of agar plate.
Line 137
12 -> Twelve
141
50 mL may be 50 μL
Lines 168 to 170
“The amplification products were electrophoresed at 90 V for 60 min in an electrophoresis chamber (BioRad Laborato ries, Inc), and visualized using the ChemiDoc Imaging System (BioRad Laboratories, Inc.), with HyperLadder 100 bp (Bioline Reagents, Ltd.) as molecular weight marker.”
Did the authors use agarose gel? If so, please indicate it.
Please also indicate how the authors visualized the DNA bands. Did they use SYBR Green?
Lines 240 to 244
2.5.5. Biogenic Amine Production PCR-Detection
The isolated DNA was subjected to PCR amplifications to detect the presence of the histidine decarboxylase (hdc), tyrosine decarboxylase (tdc) and ornithine decarboxylase (odc) genes by using the primers CL1-JV17HC, TD2-TD5, and 3–16, respectively, as previously described [39–41], PCR products were visualized as described above.
To confirm the absence of the histidine decarboxylase (hdc), tyrosine decarboxylase (tdc) and ornithine decarboxylase (odc) genes by PCR, the authors should use positive control DNA template.
Lines 272 to 273
, and one-way ANOVAs when comparing data between different isolates.
ANOVA itself does not have comparing function. Please show how the comparison analysis was performed, for example “Tukey-Kramer” and “Duncan's multiple range test (MRT)”.
Line 292
“Interestingly, the microbiota isolated from the rotifer tank inhibited ichthyopathogens” may be “Interestingly, the bacterial strains isolated from the rotifer tank inhibited ichthyopathogens”
Figure 3
Please explain * with respect to p values.
Please also indicate the number of biological replicates.
Author Response
Dear Editor,
In the revised version of the manuscript, we have introduced the corrections according to the comments raised by the reviewers. As suggested by the reviewers, we have divided the initial paragraph, which is considered by us as a fundamental for the correct contextualization of the issue, into three different paragraphs in the revised version of the manuscript, thus making it easier for the reader. Additionally, the material and methods section has been extensively shortened and streamlined in the revised version of the manuscript. However, please note that some additional information has been included in the revised version of the manuscript, as suggested by the Reviewer 1. Likewise, the results and discussion section has also been shortened and systemized in the revised version of the manuscript. Please, find below our point-by-point responses to those concerns.
REVIEWER 1:
In this article, the authors isolated 45 potential probiotic candidates from a rotifer rearing tank, examined their antimicrobial activity against several ichthyopathogens, and selected 2 Lacticaseibacillus paracasei and 2 Lactiplantibacillus plantarum strains. They further examined these 4 strains by in vitro safety assessment experiments.
The experiments were well-designed, and results contain several useful information on probiotics in aquaculture. The reviewer, however, found several concerns which should be addressed before publication as follows.
Thank you very much for your constructive and positive criticisms, comments and recommendations on this manuscript.
Major concern
The accession numbers of the 16S rRNA should be provided in “2.3. Taxonomic Identification of Selected Isolates” of the Materials and Methods section, because it is mentioned in Instructions for Authors of Animals that “Accession numbers of RNA, DNA and protein sequences used in the manuscript should be provided in the Materials and Methods section.”
The reviewer believes that this paper should not be published before the author provide the accession numbers of the 16S rRNA.
Thank you very much for your suggestion. The prokaryotic 16S rRNA gene sequences have been deposited at DDBJ/ENA/GenBank and their accession numbers have been included in the revised version of the manuscript.
Figure 3
What are the Y-axis values?
In line 252 it says “S. aureus ATCC® 25923 was included as a positive control”, but no data on S. aureus ATCC® 25923 is shown in Figure 3. The reviewer thinks it is better to express the results as the relative values to that of positive control.
Thank you for your remarks. Your concerns have been duly addressed, and the revised version of the manuscript makes it clearer, both in the text and in Figure 3 Y-axis and legend, that the results are expressed as relative (%) biofilm formation respect to those of the reference strain (S. aureus ATCC® 25923).
Minor points
119 to 121
Briefly, the 45 isolates were stabbed onto MRS agar plates and incubated at 30°C for 5 h, and then, 40 mL of the corresponding soft agar (0.8%, w/v) medium containing about 1×105 CFU/mL of the pathogen strain was poured onto the plates.
Please indicate the size of agar plate.
Thank you for your comment. For this specific assay, agar plates of 100 x 15 mm were used. However, as suggested by the Reviewer 2, description methods have been shortened and streamlined, and this paragraph has been deleted in the revised version of the manuscript.
Line 137
12 -> Twelve
Thank you for your comment. It has been accordingly changed in the revised version of the manuscript.
141
50 mL may be 50 μL
Thank you for pointing out this mistake. It has been accordingly changed in the revised version of the manuscript.
Lines 168 to 170
“The amplification products were electrophoresed at 90 V for 60 min in an electrophoresis chamber (BioRad Laboratories, Inc), and visualized using the ChemiDoc Imaging System (BioRad Laboratories, Inc.), with HyperLadder 100 bp (Bioline Reagents, Ltd.) as molecular weight marker.”
Did the authors use agarose gel? If so, please indicate it.
Please also indicate how the authors visualized the DNA bands. Did they use SYBR Green?
Thank you for your observation. An agarose (1,5% w/v) gel electrophoresis was performed in this assay, which was dyed with GelRed Nucleic Acid Gel Stain (Biotium, Inc., Fremont, California, USA) before being visualized in a ChemiDoc Imaging System (BioRad Laboratories, Inc.). This information has been included in the revised version of the manuscript.
Lines 240 to 244
2.5.5. Biogenic Amine Production PCR-Detection
The isolated DNA was subjected to PCR amplifications to detect the presence of the histidine decarboxylase (hdc), tyrosine decarboxylase (tdc) and ornithine decarboxylase (odc) genes by using the primers CL1-JV17HC, TD2-TD5, and 3–16, respectively, as previously described [39–41], PCR products were visualized as described above.
To confirm the absence of the histidine decarboxylase (hdc), tyrosine decarboxylase (tdc) and ornithine decarboxylase (odc) genes by PCR, the authors should use positive control DNA template.
Thank you for pointing this out. For biogenic amine genes detection by PCR, Lactobacillus sp. 30A and Enterococcus faecium L50 were used as positive and negative controls for was used for hdc and odc, and tdc, respectively. This information has been added in the revised version of the manuscript.
Lines 272 to 273
and one-way ANOVAs when comparing data between different isolates.
ANOVA itself does not have comparing function. Please show how the comparison analysis was performed, for example “Tukey-Kramer” and “Duncan's multiple range test (MRT)”.
Thank you for your observation, this information was lacking in the original manuscript. It has been revised and corrected in the revised version.
Line 292
“Interestingly, the microbiota isolated from the rotifer tank inhibited ichthyopathogens” may be “Interestingly, the bacterial strains isolated from the rotifer tank inhibited ichthyopathogens”
Thank you for your suggestion, it has been accepted and revised in the new version of the manuscript.
Figure 3
Please explain * with respect to p values.
Please also indicate the number of biological replicates.
Thank you for your comments. Regarding the first part of your comment, in the revised version of the manuscript you can find a new Figure 3 description, in which the p values (*) are fully explained. On the other hand, you can find the second part of your comment addressed between lines 264 and 266 of the revised version of the manuscript.
Reviewer 2 Report
Comments and Suggestions for Authors
This submitted manuscript (ID 2949802) entitled "Antimicrobial Activity, Genetic Relatedness, and Safety Assessment of Potential Probiotic Lactic Acid Bacteria, Isolated from a Rearing Tank of Rotifers (Brachionus plicatilis) Used as Live Feed in Fish Larviculture" mainly compared the antimicrobial activities of 45 isolates from three different origins in the rearing tank of rotifers and investigated the genetic relatedness based on their direct antimicrobial activity. Furthermore, in vitro safety assessment of the selected candidates strains were performed to evaluate the application potential in turbot aquaculture. Four LAB Strains, Lc. paracasei (BF3 and RT4) and Lp. plantarum (BF1 and WT12), exhibited superior data than the other strains. Results from this study could demonstrate the probiotic potential of microbiota from the rearing tank of rotifers and provide the technical support for developing new, safe, and sustainable bio-control approaches in aquaculture.
The core content of this manuscript is valuable for fish farming systems. However, the manuscript contains over-long presentation of "3.Results and Discussion" in the main text. In this case, the authors need to proofread throughout this paper and condense the length of the manuscript before it is accepted.
Major comments:
1. In "1. Introduction" section, the first paragraph is too long and contains multiple implications. It would be preferable to divide the present paragraph (Line 46-96) into 2-3 paragraphs according to their corresponding focuses.
2.The "2.Materials and Methods" section incorporates excessive detail on methods. The procedure of conventional method could be omitted and streamlined in the methods description of this section, or included in the supplemental section.
3. In Line 204-205, the labeling of the cited reference should be presented with numbers in the paper. There were similar problems in other parts. Please check the whole manuscript and delete the publication year in the revised paper.
4. Regarding the section of "3.Results and Discussion", the present discussion of the results is excessively long and repetitive. For example, the second paragraph (Line 383-430) in "3.2. In Vitro Safety Assessment of the Four Selected LAB Strains" is much too long and should be shortened or divided into multiple paragraphs. It would be preferable to streamline and reorganize the "3.Results and Discussion" with emphasis for better clarifying your findings in this study.
5. The legends of some figures were missing in this manuscript (Figure 2-3). For example, Figure 3 lacks the description of asterisk. An explanation as to what the asterisks in Figure 3 should be provided in the revised legends of figures.
6. In the section of "References", the authors should check the reference format carefully after reading the guide for authors. For example, in Reference 16, the duplicate publication year should be removed. There were similar errors in the other references. Also, it is suggested to make sure at least 50% of the references within 5 years (2019-2024).
Minor comments:
1. Many journals currently limit the maximum number of author-provided keywords, with often no more 6 keywords. There were 8 keywords included in the text (Line 43-44). Please revise the "Keywords" section to meet the restriction on the number of keywords in this journal according to the information to related guides.
2. In Line 147, replace "1,5" with "1.5".
3.Check the symbols for mass unit and volume unit in this study according to the information to related guides. For example, Line 188, please replace "ug/mL" with "μg/mL ". There were similar errors in the other part of this manuscript. Please revise accordingly.
4. In Line 494-495, the relevant information is missing. It is suggested to complete it in revised manuscript.
5. Delete the blank page (Last Page, Page 17).
Other errors were highlighted in yellow and shown in the PDF file.
Therefore, this manuscript will be reconsidered after major revision.

This manuscript (Animals ID 2949802) entitled "Antimicrobial Activity, Genetic Relatedness, and Safety Assessment of Potential Probiotic Lactic Acid Bacteria, Isolated from a Rearing Tank of Rotifers (Brachionus plicatilis) Used as Live Feed in Fish Larviculture" is well-written. But there are still some mistakes, such as the symbols for mass unit and volume unit. Also, the "Results and Discussion" section in this manuscript is excessively long, with several redundancy between the Results and the Discussion.
Author Response
Dear Editor,
In the revised version of the manuscript, we have introduced the corrections according to the comments raised by the reviewers. As suggested by the reviewers, we have divided the initial paragraph, which is considered by us as a fundamental for the correct contextualization of the issue, into three different paragraphs in the revised version of the manuscript, thus making it easier for the reader. Additionally, the material and methods section has been extensively shortened and streamlined in the revised version of the manuscript. However, please note that some additional information has been included in the revised version of the manuscript, as suggested by the Reviewer 1. Likewise, the results and discussion section has also been shortened and systemized in the revised version of the manuscript. Please, find below our point-by-point responses to those concerns.
REVIEWER 2:
This submitted manuscript (ID 2949802) entitled "Antimicrobial Activity, Genetic Relatedness, and Safety Assessment of Potential Probiotic Lactic Acid Bacteria, Isolated from a Rearing Tank of Rotifers (Brachionus plicatilis) Used as Live Feed in Fish Larviculture" mainly compared the antimicrobial activities of 45 isolates from three different origins in the rearing tank of rotifers and investigated the genetic relatedness based on their direct antimicrobial activity. Furthermore, in vitro safety assessment of the selected candidates strains were performed to evaluate the application potential in turbot aquaculture. Four LAB Strains, Lc. paracasei (BF3 and RT4) and Lp. plantarum (BF1 and WT12), exhibited superior data than the other strains. Results from this study could demonstrate the probiotic potential of microbiota from the rearing tank of rotifers and provide the technical support for developing new, safe, and sustainable bio-control approaches in aquaculture.
The core content of this manuscript is valuable for fish farming systems. However, the manuscript contains over-long presentation of "3.Results and Discussion" in the main text. In this case, the authors need to proofread throughout this paper and condense the length of the manuscript before it is accepted.
Thank you very much for your constructive and positive criticisms, comments and recommendations on this manuscript.
Major comments:
- In "1. Introduction" section, the first paragraph is too long and contains multiple implications. It would be preferable to divide the present paragraph (Line 46-96) into 2-3 paragraphs according to their corresponding focuses.
Thank you for your comment. We consider the first paragraph to be fundamental to contextualize the subject of this study. However, as suggested, we have divided this initial paragraph into three different paragraphs in the revised version of the manuscript, thus making it easier for the reader to understand the scientific context of the subject.
2.The "2.Materials and Methods" section incorporates excessive detail on methods. The procedure of conventional method could be omitted and streamlined in the methods description of this section, or included in the supplemental section.
Thank you for your recommendation. As suggested, description methods have been extensively shortened and streamlined in the revised version of the manuscript. However, please note that some additional information has been included in the revised version of the manuscript, as suggested by the Reviewer 1.
- In Line 204-205, the labeling of the cited reference should be presented with numbers in the paper. There were similar problems in other parts. Please check the whole manuscript and delete the publication year in the revised paper.
Thank you for your recommendation. Corrected as suggested, where appropriate, in the revised version of the manuscript.
- Regarding the section of “3.Results and Discussion”, the present discussion of the results is excessively long and repetitive. For example, the second paragraph (Line 383-430) in “3.2. In Vitro Safety Assessment of the Four Selected LAB Strains” is much too long and should be shortened or divided into multiple paragraphs. It would be preferable to streamline and reorganize the “3.Results and Discussion” with emphasis for better clarifying your findings in this study.
Thank you for your comments. As suggested, the section 3. Results and Discussion has been shortened, streamlined and reorganized in the revised version of the manuscript.
- The legends of some figures were missing in this manuscript (Figure 2-3). For example, Figure 3 lacks the description of asterisk. An explanation as to what the asterisks in Figure 3 should be provided in the revised legends of figures.
Thank you for your comments. The missing information has been included in the revised version of the manuscript.
- In the section of “References”, the authors should check the reference format carefully after reading the guide for authors. For example, in Reference 16, the duplicate publication year should be removed. There were similar errors in the other references. Also, it is suggested to make sure at least 50% of the references within 5 years (2019-2024).
Thank you for your comments. Regarding your first comment, these typos have been corrected accordingly throughout the text and bibliography in the revised version of the manuscript. Moreover, the bibliography has been updated, so at least 50% of the references included in the revised version of the manuscript were published between 2019 and 2024.
Minor comments:
- Many journals currently limit the maximum number of author-provided keywords, with often no more 6 keywords. There were 8 keywords included in the text (Line 43-44). Please revise the “Keywords” section to meet the restriction on the number of keywords in this journal according to the information to related guides.
Thank you for your observation. As suggested, the revised version of the manuscript only includes 6 keywords.
- In Line 147, replace “1,5” with “1.5”.
Thank you for pointing out this mistake, which has been corrected in the revised version of the manuscript.
3.Check the symbols for mass unit and volume unit in this study according to the information to related guides. For example, Line 188, please replace “ug/mL” with “μg/mL “. There were similar errors in the other part of this manuscript. Please revise accordingly.
Thank you for pointing out this mistake, which has been corrected in the revised version of the manuscript.
- In Line 494-495, the relevant information is missing. It is suggested to complete it in revised manuscript.
Thank you for your comment, which has been corrected in the revised version of the manuscript.
- Delete the blank page (Last Page, Page 17).
Thank you for your observation. This blank page has been deleted in the revised version of the manuscript.
Round 2
Reviewer 1 Report
Comments and Suggestions for Authors
In the revised manuscript, the authors provided the accession numbers of the 16S rRNA sequences, and also addressed the reviewer’s concerns. The reviewer therefore believes that the revised manuscript is now acceptable for publication in Animals.
Reviewer 2 Report
Comments and Suggestions for Authors
This revised manuscript (ID 2949802) entitled "Antimicrobial Activity, Genetic Relatedness, and Safety Assess-Ment of Potential Probiotic Lactic Acid Bacteria, Isolated from a Rearing Tank of Rotifers (Brachionus plicatilis) Used as Live Feed in Fish Larviculture" has been carefully revised as suggested. Although there was a small typo in Line 320 (replace "ere" with "were"), the current revision is suitable for acceptance.
Comments on the Quality of English Language
This revised manuscript (ID 2949802) entitled "Antimicrobial Activity, Genetic Relatedness, and Safety Assess-Ment of Potential Probiotic Lactic Acid Bacteria, Isolated from a Rearing Tank of Rotifers (Brachionus plicatilis) Used as Live Feed in Fish Larviculture" has been carefully revised as suggested. Although there was a small typo in Line 320 (replace "ere" with "were"), the current revision is suitable for acceptance.